# An Effective Method of Detecting Characteristic Points of Impedance Cardiogram Verified in the Clinical Pilot Study

**DOI:** 10.3390/s22249872

**Published:** 2022-12-15

**Authors:** Ilona Karpiel, Monika Richter-Laskowska, Daniel Feige, Adam Gacek, Aleksander Sobotnicki

**Affiliations:** 1Łukasiewicz Research Network, Institute of Medical Technology and Equipment, Roosevelt 118, 41-800 Zabrze, Poland; 2Institute of Physics, University of Silesia, 75 Pułku Piechoty 1A, 41-500 Chorzów, Poland; 3PhD School, Silesian University of Technology, 2A Akademicka, 44-100 Gliwice, Poland

**Keywords:** cardiac, cardiac function, heart, heart failure, hemodynamics, impedance cardiography, remote monitoring

## Abstract

Accurate and reliable determination of the characteristic points of the impedance cardiogram (ICG) is an important research problem with a growing range of applications in the cardiological diagnostics of patients with heart failure (HF). The shapes of the characteristic waves of the ICG signal and the temporal location of the characteristic points B, C, and X provide significant diagnostic information. On this basis, essential diagnostic cardiological parameters can be determined, such as, e.g., cardiac output (CO) or stroke volume (SV). Although the importance of this problem is obvious, we face many challenges, including noisy signals and the big variability in the morphology, which altogether make the accurate identification of the characteristic points quite difficult. The paper presents an effective method of ICG points identification intended for conducting experimental research in the field of impedance cardiography. Its effectiveness is confirmed in clinical pilot studies.

## 1. Introduction

The cardioimpedance (ICG) method is becoming more and more popular among clinical specialists as well as in the field of research and science. Searching the PubMed database for the term “cardioimpedance” we get only seven results over the period 1979–2022. On the other hand, the phrase “impedance cardiography” shows nearly 3000 results, including 156 reviews. By narrowing the area of the research solely to “ICG characteristic points” we obtain 43 references.

In 1966, Kubicek developed a practical method for monitoring stroke volume and cardiac output for NASA (National Aeronautics and Space Administration), which was used by participants in the Apollo space program [1]. In the 1980s, thanks to the work of Bernstein and Sramek’s teams, a tetrapolar technique was developed using eight electrodes placed on the side surfaces of the neck and torso, which is now commonly used in available measuring devices [1,2]. The method has been known for about 70 years, but until recently it was primarily the object of military interest. The development of computer algorithms and the miniaturization of equipment allowed for its wide application in the civil treatment [3].

Techniques have been developed to measure and monitor basic hemodynamic parameters in humans using ICG, also known as thoracic impedance plethysmography, thoracic electrical bioimpedance, or reocardiography. The ICG measurement technique, also known as being cheap and non-invasive, and the use of small currents of 2–4 mA with a frequency of 20–100 kHz, is completely imperceptible and safe. It enables the determination of stroke volume and minute capacity, as well as the assessment of factors influencing the preload (through TFC measurement) and afterload (SVR measurements, systemic vascular resistance, SVRI, systemic vascular resistance index), contractility (measurements: ACI, acceleration index, VI, velocity index), PEP, preejection period, LVET, left ventricular ejection time, STR, systolic time ratio), and heart rate. These parameters can be assessed in various ways—invasively (thanks to, e.g., direct and indirect Fick’s method, cardiac catheterization, and ventriculography) including dye dilution method, thermodilution method (PATD, PAC-CO), continuous cardiac output measurement method (PAC-CCO), Ventriculography, and Pulse Wave Analysis (PWA), Pulse Contour Analysis (PICCO), Pulse Power Analysis (LIDCO), and APCO (Vigileo), and with reduced invasiveness methods (e.g., radioisotope angiography, digital substraction angiography (DSA), and high-speed computed tomography, which require administration of a contrast agent to the patient). Non-invasive methods include magnetic resonance imaging, chest echocardiography, photoacoustic gas assessment, electrical velocimetry (EV), and the respiratory and electrical impedance mentioned above. The electrical cardiometry (EC) method is based on a modified measurement of the thoracic electrical bioimpedance (TEB).

Impedance cardiography is used in the diagnosis and treatment of patients with arterial hypertension [3,4,5,6,7,8,9] as well as heart failure. In hypertension, performing this test enables the selection of drugs that best suit the cause of the high blood pressure.

Heart failure (HF) is a significant clinical, social, and economic problem, posing a huge challenge for healthcare systems. In Poland, almost 1 million people suffer from HF, and every year 60,000 of them die because of its decompensation. This number includes patients with both reduced (HFrEF) and preserved ejection fraction (HFpEF). The extended survival of HF patients results in an increasing number of hospitalizations (Heidenreich). Despite continuous medical development and implementation of new technologies, the prognosis of patients with heart failure remains unfavorable, and the average 5-year survival rate is about 50%. The economic and social costs of heart failure are very high and have shown a steady upward trend in recent years. Heart failure accounts for 11% of all hospitalizations in Poland and is the most common cause in the population of patients over 65 years of age. The biggest problem is repeated hospitalizations within a month after discharge from the hospital. It concerns every fourth patient.

In the case of heart failure, impedance cardiography allows clinician to select drugs so that the patient takes them in optimal doses that ensure the best functioning on a daily basis. Moreover, it is a method that can reveal cardiovascular disorders at a very early stage. Especially people with elevated but still normal blood pressure may benefit from this test. Monitoring haemodynamic changes with ICG is used in intensive care units, operating theaters, and hemodialysis stations; repeated measurements bring a lot of haemodynamic information during the treatment of patients with hypertension and heart failure as well as pregnant patients with cardiological problems and pregnancy poisoning.

To meet this problem, we have created a system of care for patients with heart failure based on telemedicine tools.

The average annual cost of hospitalizing a patient with heart failure is 15 times greater than the cost of outpatient care. One of the reasons for this is the unsatisfactory level of outpatient care due to the limited access of patients with heart failure to cardiological consultations, which they often require urgently. Well-organized outpatient care is a prerequisite for reducing rehospitalizations and improving the prognosis of patients with heart failure.

From the point of view of the development of new, more accurate, effective methods and devices for ICG impedance cardiography systems, the most important problem is the analysis of the ICG signal, which leads to the determination of characteristic points B, C, and X related to important events of the heart cycle [10,11,12,13,14]. Point B is often difficult to spot. Finding a point is sometimes error-prone because it is sometimes blurry in the record or difficult to detect due to artifacts [15]. There is a gradual development of technology based on machine learning techniques. Immediately during and after the pandemic, papers were published in which artificial intelligence methods were successfully used to verify and classify the ICG signal [16,17] or the innovative approach of the pattern recognition artificial neural network (PRANN)) to identify five different ICG complex waveforms [16]. Along with the solutions of automated analysis or classification [18,19,20], there are also tools that can support through visualization (https://physiodatatoolbox.leidenuniv.nl, accessed on 1 December 2022), which gives hope for the emergence of new solutions in the form of medical devices.

Telemedicine, which stands for the remote delivery of healthcare services using information and telecommunications technology [21] can also be successfully used for ICG monitoring. It includes many forms of service provision, of which the most popular are teleconsultation and telemonitoring, and the pandemic period has shown that such solutions are necessary to improve the quality of medical services. Recently, in the times of the pandemic, modern solutions such as e-dismissal or e-prescription have become a modern tool to take care of patients. In addition, more technologically advanced services have gained an unquestionably strong position in the healthcare system as an improvement of classical medical activities.

## 2. Materials and Method

### 2.1. Bioimpedance Cardiography Method

The bioimpedance cardiography method is a diagnostic method allowing for the continuous and real-time monitoring of the hemodynamic parameters. It consists of the application of a low-amplitude alternating current to the thorax which can be regarded as a cylindric volume conductor. The transient decrease (increase) in intrathoracic blood volume during systole (diastole) observed during each cardiac cycle results in the variations of the measured bioimpedance ΔZ.

### 2.2. Hemodynamic Parameter Calculation

In the real circulatory system, the pulse wave spreads continuously. At the beginning of the ejection phase, there is an increased inflow of blood. It decreases over time and the outflow starts to dominate over the inflow. This is the reason why the maximum value of impedance $Z_{\max}$ is underestimated. Therefore, in order to determine the real value of $Z_{\max}$, one applies the extrapolation of the slope in the obtained signal, assuming that the inflow of the blood has a dominant influence on the shape of the cardioimpendance cycle. In the Figure 1 we present the simultaneously recorded ECG, aortic pressure $P_{\mathrm{aorta}}$, REO (bioimpedance *Z*), and $\frac{dZ}{dt}$ (ICG) signals. The figure also illustrates how to determine the real value of the change in the bioimpedance ΔZ.

Based on Figure 1, by applying simple mathematical transformations, we obtain the following formula for the ΔZ:(1)$$\Delta Z={\left(\frac{dZ}{dt}\right)}_{max}\cdot VET,$$
where:${\left(\frac{dZ}{dt}\right)}_{max}$ denotes the amplitude of the systolic wave defined as the difference between values of the B and C points marked on the ICG signal;VET stands for the ventricular ejection time associated with the time between the occurrence of the B and X points.

Then, the stroke volume (SV) defined as the volume of blood pumped from the ventricle per beat can be calculated according to the *Sramek–Bernstein* equation  [22,23]:(2)$$SV_{SB}=\frac{W}{IW}\cdot \frac{{\left(0.17\cdot H\right)}^{3}}{4.25\cdot Z_{0}}\cdot {\left(\frac{dZ}{dt}\right)}_{max}\cdot VET$$
where:W[kg] and H[cm] is the weight and height of a subject;$(\frac{dZ}{dt}{)}_{max}$ is the peak amplitude of the ICG signal;VET denotes the ventricular ejection time.

Parameter IW stands for the ideal weight calculated according to the following formula  [24]:(3)$$IW=\left\{\begin{array}{cc} 0.534\cdot H-17.36 & \mathrm{for}\;\mathrm{man},\\ 0.534\cdot H-27.36 & \mathrm{for}\;\mathrm{woman}. \end{array}\right.$$

The determination of the stroke volume allows for the estimation of another important hemodynamic parameter called cardiac output (CO) defined as the amount of blood pumped by the heart per minute:(4)$$CO=SV_{SB}\times HR,$$
where HR is the heart rate.

From Equation (Equation 2) one can easily infer that in order to calculate the values of the stroke volume and cardiac output, we first need to find the values of the peak amplitude of the ICG signal ${\left(\frac{dZ}{dt}\right)}_{max}$ and ventricular ejection time VET. They are, however, dependent on the positions of characteristic points B, C, and X marked in the lowest part of the Figure 1. Their physiological significance and characteristics in the ICG, ECG, and REO signals are presented in Table 1.

### 2.3. The Algorithm of Identification of the Characteristic Points Implemented in the 4hearts AMULET System

Based on the patent (No. 418977), an algorithm was created to identify feature points, which was implemented in an application and a medical device developed in the Łukasiewicz Research Network, Institute of Medical Technology and Equipment in Zabrze. The proposed algorithm of the identification of the characteristic points in the ICG $\left(\frac{dZ}{dt}\right)$ signal consists of its analysis in the specified time window T. The length of this window and its position is dependent on the previously determined positions of the fiducial points in the ECG signal and is set as:(5)$$T=[R-\frac{7}{20}\cdot \langle RR\rangle ,R+\frac{13}{20}\cdot \langle RR\rangle ],$$
where 〈RR〉 denotes the mean duration of the distance between two consecutive R points, while R stands for the current position of the R point in the current cycle. The way the fiducial points Q, R, S, and T occur in ECG is identified and introduced in Refs.  [25,26,27,28].

The C point on ICG is identified as the point characterized by the maximum amplitude in a given time window T.

The B point on ICG always occurs before the C point and can be identified as the first positive apex occurring after the S-wave in the second derivative of the ICG signal (REO_3). As can be inferred from Figure 2, it always occurs after the QRS complex and before the beginning of the T-wave. Figure 2 presents the time relationships between the basic rheogram (REO), its first derivative, i.e., the ICG reocardiogram and the ECG electrocardiogram record, as well as the location of the characteristic points. The shape of the ICG reocardiogram complexes is very diverse, which is why their analysis in terms of determining the characteristic points is difficult, especially in the presence of interference. Considering the availability of the ECG recorded simultaneously with the rheographic recording, the possibility of finding characteristic points of the ICG reocardiogram using previously determined ECG characteristic points should be used. The need to use the ECG electrocardiogram in the analysis of the ICG reocardiogram occurs in the case of arrhythmias that can be recognized on the ECG record, but on the ICG reocardiogram itself, they make the detection of the syndrome extremely difficult or even impossible. The developed algorithm boils down to the analysis of the ICG signal in a specific time window. The length of the window and its location on the time axis depends on the previously defined location of characteristic points in the PQRST complex of the ECG signal. The time window includes a fragment of the ICG signal, which is the so-called region of interest, which is used to analyze the ICG signal. In this case, the ECG signal is an auxiliary signal that synchronizes and controls the analysis of the ICG signal. The authors developed original decision criteria for the numerical identification of characteristic waves on the waveforms of the ECG and ICG signal and its derivatives, enabling the determination of characteristic points B, C, and X.

The X point on ICG is associated with the first minimum occurring in the ICG signal immediately after the end of the T-wave. In the case the T-wave can not be identified in the corresponding ECG signal, the X point is established as the first minimum in ICG, appearing after the maximum of the systolic wave REO after the C point.

The general working principle of the above-presented algorithm is illustrated in Figure 2, where the positions of these points are marked in relation to the pressure signals registered during one cardiac cycle in the pulmonary artery (PA), right atrium (RA), and right ventricle (RV). While ECG, REO, and PA were recorded simultaneously, RV and RA signals had to be “superimposed” on the remaining signals due to the impossibility of simultaneous recording of pressures in the pulmonary artery, right ventricle, and right atrium. The ICG signal was obtained by the numerical differentiation of the REO according to the formula: (6)$$\mathrm{REO}\_1=\mathrm{ICG}\left[n\right]=0.1f_{s}\cdot \left(2\cdot \mathrm{REO}[n+2]+\mathrm{REO}[n+1]-\mathrm{REO}[n-1]-2\cdot \mathrm{REO}[n-2]\right),$$
where $f_{s}=500\;\mathrm{Hz}$ is the sampling frequency during the measurement. REO[n] is a waveform obtained by digitizing the recorded analog cardiography impedance signal.

### 2.4. The 4hearts AMULET Software

The above-described method of the characteristic points identification and determination of the hemodynamic parameters from the ICG signal was implemented in the modern software 4hearts AMULET. Its working principle is depicted in Figure 3. The method of determination of the characteristic points applied in this system is presented in Section 2.3.

The AMULET software is designed for the acquisition, visualization, and archiving of the data coming from the examinations of different patients. It allows on-screen presentation of the electrocardiogram and cardiography impedance during the study, the real-time monitoring of the positions of the fiducial points and the hemodynamic parameters such as

Heart rate (HR);Stroke volume (SV) (defined in Equation (Equation 2));Cardiac output (CO) (defined in Equation (Equation 4));Pre-ejection period (PEP) defined as the time between electrical systole (Q point in ECG) and the opening of the aortic valve (B point in ICG);Ventricular ejection time (VET) associated with the time interval between B and X point;Thoracic fluid content (TFC) indicates the total fluid volume of the thorax and is derived from the inverse of the base impedance ($1/Z_{0}$).

Moreover, it enables saving the recorded examinations to the database, allowing to replay the conducted study for a more accurate analysis at a later time.

The graphical user interface (GUI) is easy and user-friendly—it allows for easy adaptation of the software to the user’s requirements (e.g., it is possible to adjust the test display format to 1, 2, or 3 channels, change the settings of the monitor, sound, and choice of the color scheme). The program control window of the developed software is depicted in Figure 4.

As can be noticed, it consists of three main parts. The upper part presents the current electrocardiogram along with the ensemble average of the signal acquired from 15 previous cardiac cycles. The middle part illustrates the waveform of the ICG signal, with the points B, C, and X marked in the positions found by the algorithm. At the bottom part of the window, one can observe the behavior of the baseline impedance $Z_{0}$ during the conducted study.

The right panel of the window present trends and instantaneous values of the hemodynamic parameters.

Currently, the developed system has a wide application in testing subjects under conditions of ischemic hypoxia and orthostatic stress caused by simulated overloads [29,30].

## 3. Results and Discussion

In order to verify the efficiency of the developed algorithm, we carried out the clinical experiments at the Silesian Center for Heart Diseases in Zabrze on 25 patients (15 women and 10 men, weight 76.88±14.5, height: 155–187 cm). ECG and ICG signals, pressures in the right heart (RA, RV) and pulmonary artery (PA) were acquired from the patients under examination.

Subsequently, the positions of the characteristic points in the ECG as well in the ICG signals were identified by a medical expert—a cardiologist in the field of bioimpedance measurements. In this manner, the database composed of the 25 records collected from the patients with diverse degrees of cardiac disease was created and was later the basis for the verification of the algorithm described in Section 2.3. For each of the 25 patients, an expert assessed exactly 15 s of recording, which corresponds on average to 18 heart cycles (studies were performed at rest).

In order to assess its effectiveness we firstly compared the positions of B, C, and X points found by the algorithm $P_{\mathrm{alg}}$ with those indicated by a medical expert $P_{\exp}$. In this aim, based on the method described in Ref. [31], we calculated for each patient ΔP (where P={B,C,X}) the mean absolute values of the differences between $P_{\exp}$ and $P_{\mathrm{alg}}$:(7)$$\Delta P=|P_{alg}-P_{exp}|$$
and their corresponding mean standard deviations σ(ΔP).

We then similarly compared the values of ventricular ejection time (VET) and the amplitude of the systolic wave ($A_{max}={\left(\frac{dZ}{dt}\right)}_{max}$) which are necessary to calculate the values of the stroke volume $SV_{SB}$ and cardiac output (see Equations (Equation 2) and (Equation 4)). These results, specified for each of the analyzed patients, are presented in Table 2.

The difference in positions of the characteristic points ΔB, ΔC, ΔX was the greatest for $\Delta X_{t}$ = 16 and for $\Delta B_{t}$ = 50 and $\Delta C_{t}$ = 4, respectively. The analysis showed that the smallest standard deviations of the differences in feature point detection times ($\sigma B_{t}$, $\sigma C_{t}$, $\sigma X_{t}$ ) were observed for $\sigma C_{t}$.

From Table 2, it can be seen that some records have significant errors in the determination of mainly B-points, less frequently X-points. These result in unacceptably large errors in the calculation of systolic wave amplitude $\Delta {\left(dZ/dt\right)}_{max}$ and/or ventricular ejection time VET, which affect the accuracy of SV stroke volume determination. Errors in the determination of B and X points occur mainly when pathological recordings with morphology significantly different from the basic shape of the cardioimpedance complex are analyzed.

Figure 5 shows three histograms that presented dispersion of mean absolute differences (ΔB, ΔC, ΔX) in the detection times of the B, C, and X points, respectively. The most remarkable noticeable differences in the presented histograms occur for point B. The results are as expected, considering that it was a group of patients that all qualified for a heart transplant. The signal in such cases is more difficult to interpret and this poses a challenge for a wider group of scientists to develop precise verification tools. In turn, box charts (Figure 6a) show two approaches, both for the points set by the expert and the algorithm proposed by us. The bioimpedance ΔZ value was presented, which clearly indicates that the results obtained for both the points designated by the expert and our algorithm are at a similar level. A slightly higher median was obtained for the algorithm proposed by us. The results showed that in both cases amplitudes (Figure 6c) are comparable, while the median is slightly different, which is directly related to the previously discussed point B. However, the analyzes showed that the algorithm proposed by us is more accurate, considering the analyzed VET value (Figure 6b). Medicine aims to improve the quality, speed, and accuracy in the presentation of results, which confirms that the work to propose a new algorithm was right. Figure 7 presents the detection error for ICG fiducial points, where, as expected, the greatest dispersion was obtained for point B.

In Figure 8 we present a scatter plot and the corresponding regression line for the relationship between the cardiac output (CO) (based on the input from an expert)and the CO for our new algorithm.

We find a high correlation (correlation coefficient r=0.96) between the results obtained from the annotations made by an expert and by our algorithm. Also the Bland–Altman plot (Figure 9) suggests a high agreement between the two approaches. In addition, to validate the effectiveness of the proposed procedure, we also performed the paired *t*-test between $CO_{expert}$ and $CO_{alg}$. Before proceeding, we checked if all necessary conditions are met to perform such a statistical test. In this aim, we verified that there are no relevant outliers present in our dataset. Additionally, we found out that according to the Shapiro–Wilk test [32] the results are indeed normally distributed (*p*-values equal to 0.37 and 0.16 for values of cardiac output as assessed by an expert and the algorithm, respectively). It implies that the application of the paired *t*-test is well justified. In fact, it revealed that the differences obtained from the two methods are statistically significant (*p*-value = 0.0006 < 0.05). We believe, however, that in most cases they do not influence the diagnosis and have no relevant clinical implications.

In order to substantiate the above statement, we additionally plot the histogram illustrating the percentage error (PE) between two methods applied (Figure 10). According to Refs. [33,34] the admissible PE is equal to 30%. As can be inferred from Figure 10 the majority of cases remain with this limit. For those that do not, some more precise techniques of the CO assessment such as Fick method or thermodilution are required.

In our view, although the above-presented algorithm cannot always replace the invasive techniques, it can be successfully applied during outpatient or telemedical treatment (for which the telemonitoring system AMULET was created for). From a clinical point of view, the bioimpedance method will not replace classic, invasive methods of measuring cardiac output parameters, e.g., thermodilution or Fick’s approach, in cases where it is necessary to determine them with high accuracy and at the same time under invasive surgical conditions involving the need for cardiac catheterization. However, in ambulatory or telemedicine applications, the bioimpedance method, even with lower accuracy of measurements, is often the only one that can be used, also offering the possibility of continuous beat-to-beat control of the stroke volume of the heart SV, which was considered in Ref. [35].

The shape of ICG varies greatly, so their analysis of characteristic points is difficult, especially in the presence of interference. Taking into account the availability of ECG recorded simultaneously with the bioimpedance signal, it takes advantage of the possibility of finding the characteristic points of ICG reocardiogram by using previously determined ECG characteristic points. The need for the use of ECG for the analysis of the ICG reocardiogram occurs in the case of arrhythmias, which can be diagnosed on the ECG, while based on the ICG signal alone makes the detection of the syndrome extremely difficult or even impossible.

It should be mentioned that finding the reliable locations of the B, C, and X points is sometimes problematic, as described more extensively in the literature [24,36,37,38,39]. The present results indicate that the B-point of the ICG signal, corresponding to the opening of the aortic valve, is highly related to the time of peak dZ/dt function. In 2016, a paper was published, where the authors found unexpected and dramatic differences in the accuracy of three algorithms, with one based on the third derivative of the impedance cardiogram performing significantly better [40]. The algorithm based on the third derivative showed the highest accuracy rate (78.77%), so it seems to be the best.

### Limitations

The presented analysis focuses on the comparison of a world expert in the field of ICG analysis with the algorithm underlying the operation of a medical device based on a patent application. There are already many works comparing the algorithms and the topic would be suitable for another review, so we will not describe them. We herein focused on presenting our solution, which is successfully used in clinical practice. In earlier works, no statistical differences were observed between the analysis for the two experts. Questions arise as to how much and to what extent the results can differ in order to obtain reliable values, which are the basis for the diagnosis. It seems that this information should be systematized, which would also make it easier to compare the results between each other and between different medical companies producing devices. This is another factor that has a huge impact on the final result, but we must take into account that companies do not provide all of the information, in particular regarding technical matters. The device (4hearts AMULET), which is based on the described algorithms, was compared with a device from another company during clinical trials. The tests were performed with two devices (for technical reasons, not at the same time)-test “after” test. No statistically significant differences were noted. The database we had at our disposal may also have an impact on the obtained results. It is also instructive to tackle the problem of the presence of only one expert in the performed analysis. Although the annotations made by two experts in the field can be different [41] it has been demonstrated in Refs. [40,42] that the potential discrepancies are not statistically relevant and have a small impact on the comparative analysis of the two procedures.

It is also important to mention the proper placement of electrodes, both ECG and ICG, which the medical staff may not always remember. The electrodes are attached in an imprecise manner, which may affect the quality of the recorded signal. The devices we use to record the signal may also differ from each other due to the various algorithms used. In a 2016 paper, Javier Rodriguez Arbol et al. [40] presented the percentage statistics describing the methodology used. It turns out that 72 % do not mention what algorithm was used, which leads to a lack of reproducibility and creates problems when comparing the results with each other. It is worth mentioning here the available device such as the MEDIS-Niccomo or the tools supporting analyzes such as ANSLAB (http://anslab.net/static/help26/impedancecardiography.html, accessed on 2 November 2022), BIOPAC (https://www.biopac.com/application/icg-impedance-cardiographycardiac-output/, accessed on 2 November 2022), or MindWare (https://support.mindwaretech.com/, accessed on 2 November 2022). In addition to the previously mentioned technical aspects, we come to issues related to individual variability and repeatability of activities. The first comparisons were made in 1993 by Dębski et al. [43], where the mean value of five physiological indices requiring B-point estimation was compared. The topic was addressed and analyzed in the course of cardiac impedance by Sherwood et al. [44], who modified the classic description of point B as a “notch”, defining it as “the beginning of a sharp drop in dZ/dt as it increases to a peak value”. Alternatively, Lozano et al. [37] proposed an indirect polynomial method that does not require the identification of the B point to estimate PEP, but rather relies on the correlation between the PEP and the time interval between the R peak on the ECG and the C point on the ICG (both easily identifiable on their respective charts). However, Van Lien, Schutte, Meijer, and de Geus [45] found significant discrepancies in different experimental conditions between the PEP estimated by the latter method and the traditional method. They concluded that even though it is a helpful tool, it should not be used as a substitute for real PEP obtained by detecting aortic valve opening. The most important conclusion from the presented comparisons is the fact that a certain degree of variability of the shape of the dZ/dt wave regarding the area of point B is noted in the literature, and a detailed description of this variability is usually omitted.

A recent review shows that the bioimpedance method, along with the methods used, is becoming more common [46]. The method of determining points, despite the variety of algorithms used, allows to obtain consistent results. The analysis of the available methods unequivocally allows us to conclude that the algorithms, based on the location of characteristic points B, C, and X, using derivatives of the bioimpedance signal, are accurate enough to be successfully implemented and used in medical diagnostics.

## 4. Conclusions

An ICG signal analysis algorithm has been developed, which enables easy determination of the location of the characteristic points B, C, and X on the amplitude-time plane and the specification of quantitative parameters. The analysis of the results obtained with our algorithm shows a high agreement of the calculated SV value in relation to the one calculated on the basis of the expert’s analysis. Nevertheless, the algorithm requires development of signal analysis in cases of arrhythmias. It is also necessary to supplement the experiments in order to obtain the SV stroke volumes with the reference methods.

## 5. Patents

Method of acquisition and processing of bioimpedance signals using a bioimpedance signal acquisition module “Application No. 418977”.

## Figures and Tables

**Figure 1 sensors-22-09872-f001:**
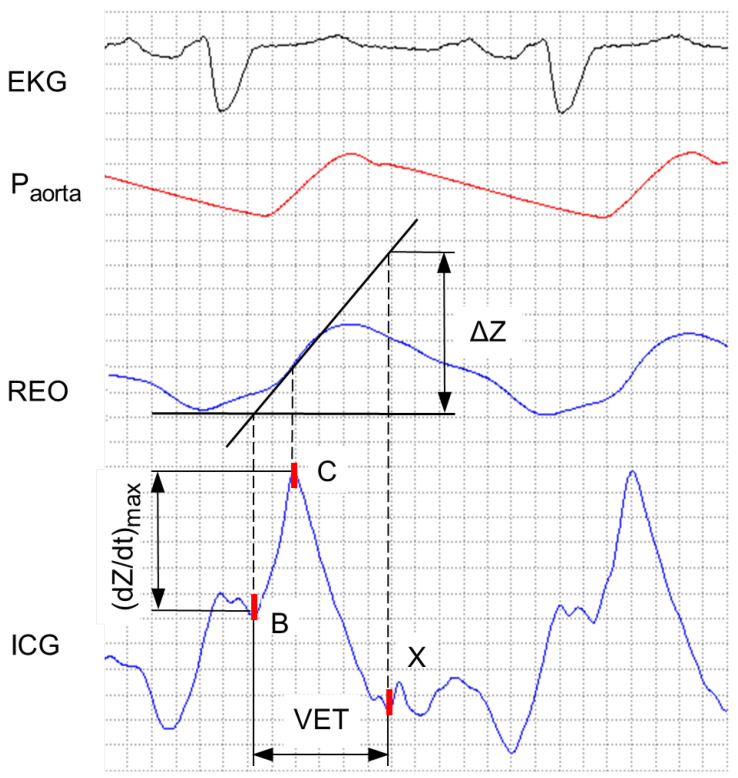
The simultaneously recorded ECG, aortic pressure $P_{\mathrm{aorta}}$, REO (bioimpedance *Z*), and ICG signals $\frac{dZ}{dt}$ signals. The line on the REO signal illustrates the way to determine the real value of the impedance variation ΔZ with a slope extrapolation. The corresponding fragment of the maximum increase of the bioimpedance ${\left(\frac{dZ}{dt}\right)}_{\max}$ found with this method is also depicted in the ICG signal. Points B, C, and X indicate the characteristic moments of the cardiac cycle and are explained in more detail in Table 1. The quantity VET illustrates the ventricular ejection time defined as the distance between X and B points.

**Figure 2 sensors-22-09872-f002:**
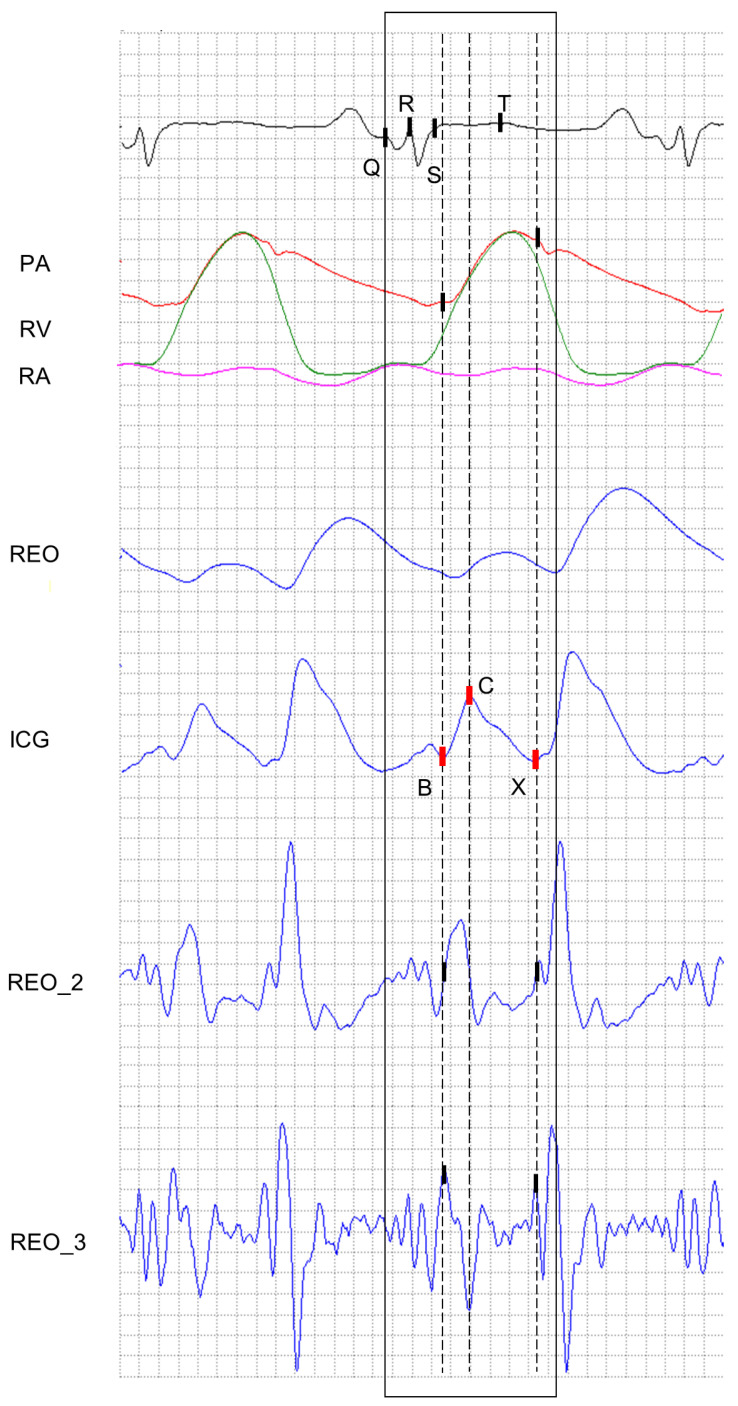
Example of signals registered during one cardiac cycle. The first signal from above illustrates the ECG signal along with its characteristic points: Q, R, S, and T. The next picture presents the pressures in the pulmonary artery (PA), right atrium (RA), and right ventricle (RV). The following waveform depicts the cardiography impedance signal and its corresponding derivatives: $\frac{dZ}{dt}$ (ICG), $\frac{d^{2}Z}{dt^{2}}$ (REO_2) and $\frac{d^{3}Z}{dt^{3}}$ (REO_3). The rectangle denotes the time window used during the identification of the characteristic points in the ICG signal.

**Figure 3 sensors-22-09872-f003:**
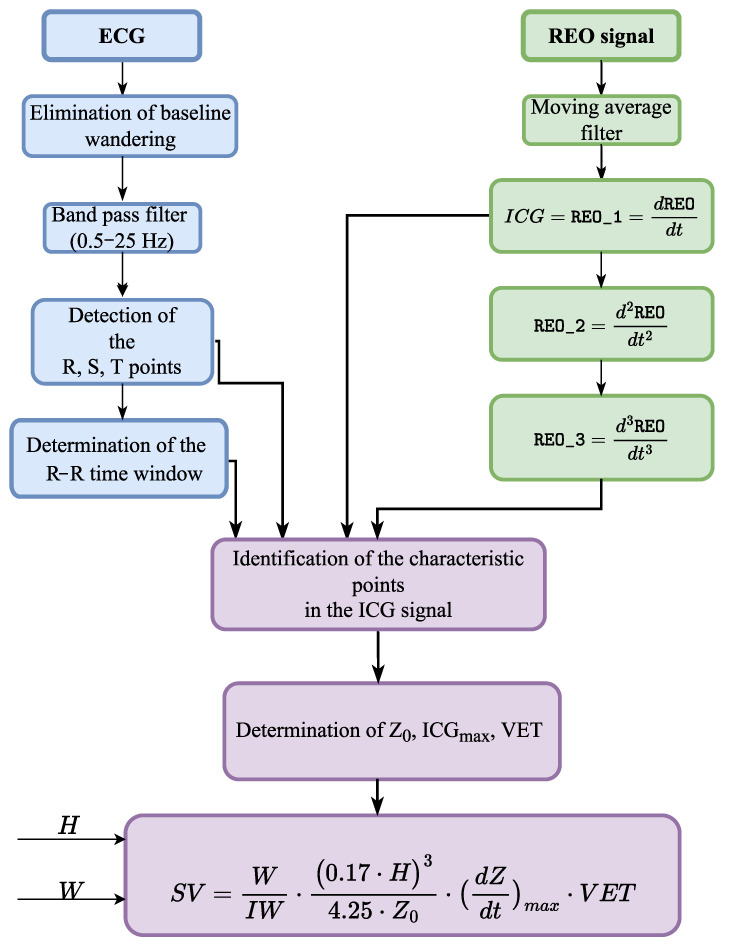
Sketch of an algorithm detecting fiducial points B, C, and X in the ICG signal within one cardiac cycle.

**Figure 4 sensors-22-09872-f004:**
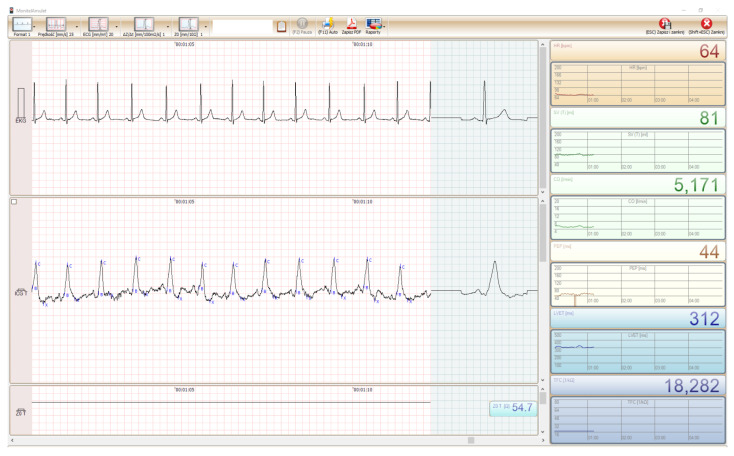
Screen of the graphic user interface of 4hearts AMULET.

**Figure 5 sensors-22-09872-f005:**
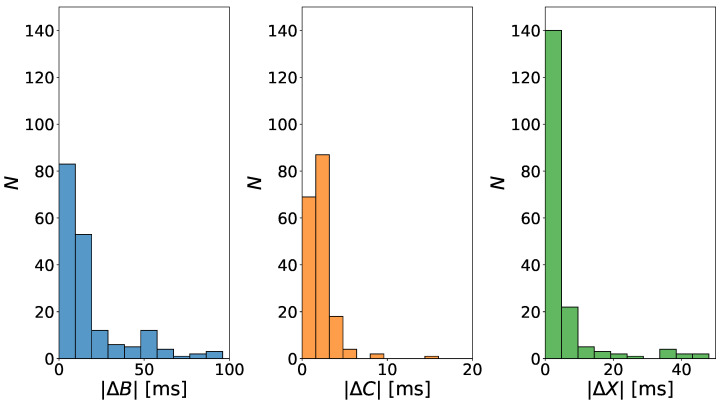
The histograms illustrating the dispersion of mean absolute differences |ΔB|, |ΔC|, |ΔX| in the detection times of the B, C, and X points, respectively. The number *N* stands for the number of observations corresponding to the presented time intervals.

**Figure 6 sensors-22-09872-f006:**
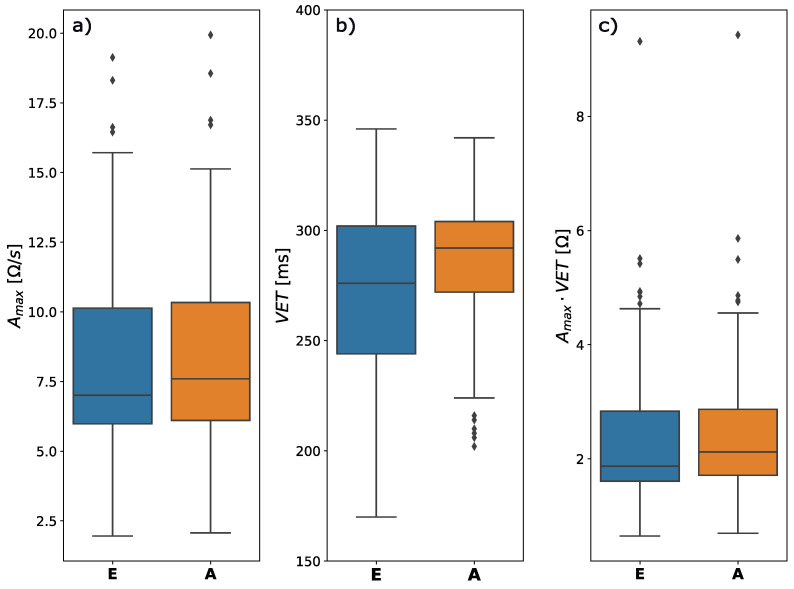
Boxplot of comparison of the two approaches: expert “E” and our algorithm “A”, consisting of three parts: I-Amax, II-VET [ms], and III-Amax [Ω].

**Figure 7 sensors-22-09872-f007:**
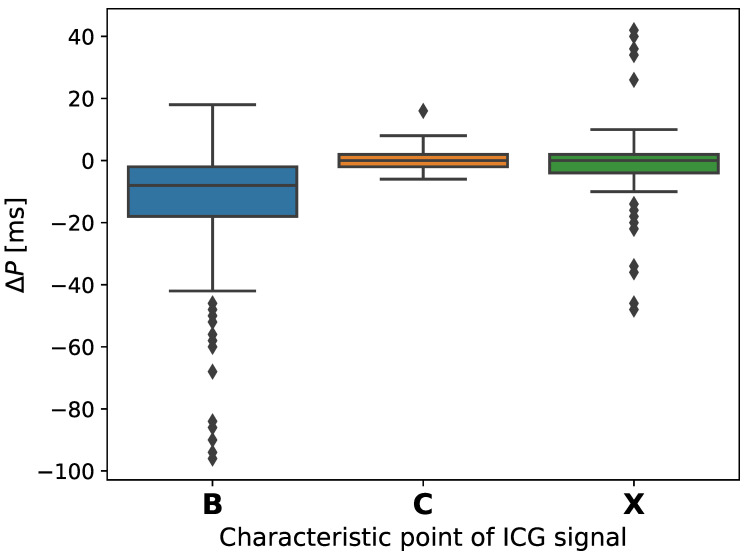
Detection error (ΔP [ms]) for the ICG fiducial points.

**Figure 8 sensors-22-09872-f008:**
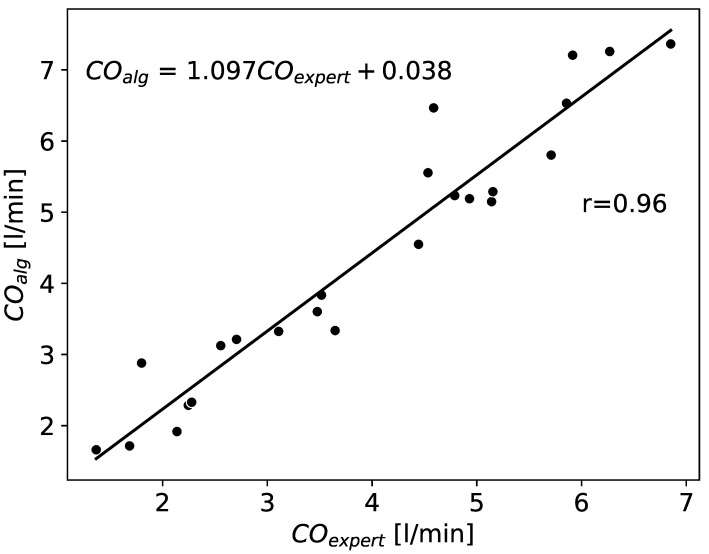
The plot illustrating the fit of the cardiac output (CO) as predicted by the algorithm ($CO_{alg}$) and resulting from the estimations made by an expert ($CO_{exp}$). The presented equation illustrates the relationship between these two approaches obtained by the method of linear regression. The number *r* stands for the Pearson correlation coefficient.

**Figure 9 sensors-22-09872-f009:**
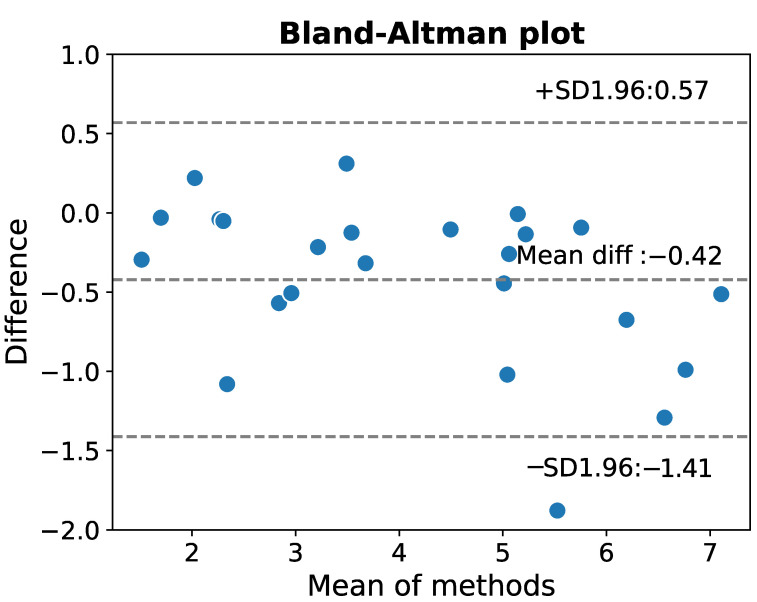
The Bland–Altman difference plot. Here, a 95% confidence interval is assumed for the means of the differences.

**Figure 10 sensors-22-09872-f010:**
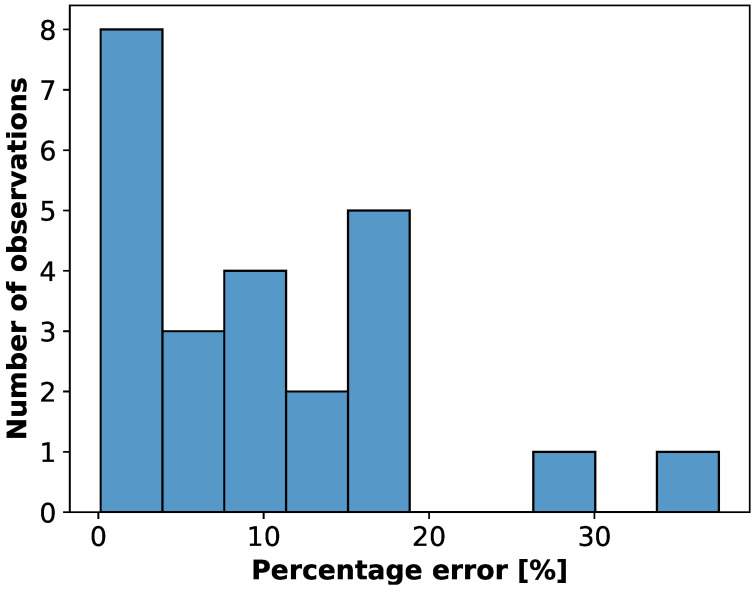
The percentage error illustrating the bias between the estimation of the cardiac output (CO) as predicted by an expert and our algorithm.

**Table 1 sensors-22-09872-t001:** The characteristic points in the ICG signal, their physiological relevance and reference to ECG and REO.

Fiducial Point	Physiological Significance	ICG Signal Characteristics	ECG and REO Signal Characteristics
**B**	end of the isometric contraction onset of the rapid ventricular ejection phase beginning of the opening of semilunar valves	beginning of the systolic wave maximum increase in blood flow acceleration	always after the QRS complex before the beginning of the T wave
**C**	represents the maximum aortic flow end of the rapid ejection phase beginning of slow ejection phase	maximum of the systolic wave	between the end of the QRS complex and the maximum of the REO systolic wave
**X**	end of ventricular contraction beginning of the closure of semilunar valves	minimum after the systolic wave, (location depends on the heart rate) maximum increase in the flow acceleration	at the end or after the T-wave

**Table 2 sensors-22-09872-t002:** The effectiveness of the proposed algorithm in identification of the characteristic B, C, and X points in the impedance cardiogram. ΔB, ΔC, ΔX denote the mean absolute values of the differences in the positions of B, C, and X points and their corresponding standard deviations σB, σC and σX (both expressed in ms) between values obtained from the algorithm and indicated by an expert. The analogous quantities ΔVET, σ(ΔVET) were also presented for VET (ventricular ejection time) and $A_{max}={\left(\frac{dZ}{dt}\right)}_{max}\;$ expressed in Ω/s (the amplitude of the systolic wave on the impedance cardiogram).

N	ΔB	σ(ΔB)	ΔC	σ(ΔC)	ΔX	σ(ΔX)	ΔVET	σ(ΔVET)	$\Delta A_{\max}$	$\sigma (\Delta A_{\max})$
1	6	4	2	2	12	20	8	10	0.37	0.33
2	18	12	2	2	4	4	10	6	0.74	0.3
3	10	6	2	2	4	2	4	4	0.32	0.28
f4	6	4	4	2	4	4	4	2	0.42	0.27
5	2	2	2	2	4	4	2	2	0.03	0.03
f6	6	6	2	2	4	4	4	2	0.09	0.08
f7	4	4	2	4	4	2	2	2	0.07	0.05
8	12	6	2	2	4	4	4	2	0.84	0.42
9	18	2	2	0	2	2	10	2	2.32	0.56
10	50	2	0	0	2	0	26	2	1.14	0.13
11	2	2	2	2	10	14	6	8	0.05	0.03
12	36	2	2	2	2	2	18	2	0.88	0.2
13	22	28	2	2	0	0	12	14	0.05	0.06
14	14	10	0	0	2	2	8	6	0.12	0.07
15	16	10	2	2	2	2	8	4	0.5	0.3
16	2	2	2	2	2	2	2	2	0.01	0.16
17	42	2	2	2	2	2	22	2	0.57	0.17
18	22	6	4	4	4	2	10	2	0.4	0.14
19	2	2	2	2	2	2	2	2	0.07	0.16
20	14	6	2	2	4	2	6	4	0.6	0.18
21	8	6	2	2	16	16	6	6	0.26	0.13
22	12	6	2	2	4	2	4	2	0.35	0.2
23	48	42	2	2	12	8	22	18	0.27	0.2
24	20	4	2	2	2	2	10	2	0.75	0.18
25	56	4	4	4	14	18	36	10	1.05	0.35

## Data Availability

Data available on request due to restrictions privacy.

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
