# Peer review of "An Effective Method of Detecting Characteristic Points of Impedance Cardiogram Verified in the Clinical Pilot Study"

_sensors, 2022, doi:10.3390/s22249872_

Round 1

Reviewer 1 Report

Summary:

The authors present a heuristic method to detect characteristic points of an ICG signal based in part on ECG and REO (bioimpedance) signals.

General concept comments: 

While the work is scientifically sound with appropriate study design and validation, the article needs to make it clear that they need ECG signal for accurate determination of the characteristic points (B, C, X as shown in Figure 3), I suggest the authors revise the title, abstract, introduction, and conclusion sections to reflect this. It is also not clear how many cardiac cycles were annotated within each of the 25 patients by the medical expert (line 195-197). The authors should provide further details about the qualifications of the medical expert (for example, is it a certified technician or a cardiologist). If practically feasible, the authors should consider annotating the same data using another independent medical expert and when there is discrepancy between the two annotators, they should have a 3rd expert resolve the differences to establish the ground truth. If this is not possible, the authors may discuss intrinsic human bias and errors in annotations as part of the limitations of the study. Due to these reasons, I recommend accepting the article after the comments are satisfactorily addressed.

Author Response

Thank you for comments.

Information on the ECG dependence, and the need to use the ECG signal has been added and we have expanded the work taken into account this information (section 2.3).

The authors supplemented the information about the specialist who was the evaluating expert and supplemented the information on the signal range used for analysis.

Unfortunately, due to the short time given to answer your comments, we were not able to make an analysis with the second expert involved. We are also not convinced if the addition of the estimations made by more experts would substantially influence the statistical comparison between annotations provided by a cardiologist and the algorithm. In fact, it has been demonstrated in [1, 2] that the annotations of the B point (which is the considered as the most difficult ICG point to identify) performed by two scorers are in good agreement and the resulting discrepancies between them are not statistically relevant. We discuss this aspect in the manuscript.

[1] Árbol, Javier Rodríguez, et al. "Mathematical detection of aortic valve opening (B point) in impedance cardiography: A comparison of three popular algorithms." Psychophysiology 54.3 (2017): 350-357.

[2] KELSEY, ROBERT M., et al. “The Ensemble-Averaged Impedance Cardiogram: An Evaluation of Scoring Methods and Interrater Reliability.” Psychophysiology, vol. 35, no. 3, 1998, pp. 337–340., doi:10.1017/S0048577298001310

Reviewer 2 Report

The paper reports on an algorithm for detecting characteristic points of impedance cardiograms. The topic is of great interest; however, some concerns need to be addressed.

1)      The first concern is about the choice of the gold standard to validate the method. In fact, as stated by the Authors "The human eye can be fallible and a refined algorithm allows you to minimize the risk of errors". Maybe, at least, the evaluation by more than one expert should be considered; only the metrics with a correlation coefficient higher than 0.9 should be included, and the mean value of the metrics evaluated by the two or more experts should be examined in further analysis. Please add this kind of analysis or discuss this aspect in the manuscript.

2)      The second concern is about the statistical analysis to validate the procedure. In order to validate a new procedure, the following analyses should be performed: a paired t-test between the gold standard and the new method to assess a bias between the two procedures; a correlation analysis between the gold standard and the new method (correlation coefficient); the equation of the linear regression (the slope of the fit should be around 1 and the constant term around 0); and a Bland-Altman plot (to evaluate the agreement between the two procedures and the presence of systematic errors in the estimation of the metric). Please add these analyses to the manuscript.

3)      Please check the language. Some sentences are very short and need to be paraphrased.

Author Response

  1. The first concern is about the choice of the gold standard to validate the method. In fact, as stated by the Authors "The human eye can be fallible and a refined algorithm allows you to minimize the risk of errors". Maybe, at least, the evaluation by more than one expert should be considered; only the metrics with a correlation coefficient higher than 0.9 should be included, and the mean value of the metrics evaluated by the two or more experts should be examined in further analysis. Please add this kind of analysis or discuss this aspect in the manuscript.

Dear Reviewer,

Thank you for this valuable comment. After reconsideration, we decided to remove the sentence you cite, i.e. "The human eye can be fallible and a refined algorithm allows you to minimize the risk of errors” from the manuscript. The truth is that the eventual errors made by a medical expert have no significant effect on the statistical comparison between the proposed algorithm and the gold standard. In fact, it has been demonstrated in [1, 2] that the annotations of the B point (which is the considered as the most difficult ICG point to identify) performed by two scorers are in good agreement and the resulting discrepancies are not statistically relevant.  Therefore, we decided to not include such analysis in the test but discuss this aspect in the manuscript.

We have introduced a new section to address your comments. In Addition, we modified section 2.3 and added point 3.1 „Limitations”. These points are an attempt to answer and presenting our position.

[1] Árbol, Javier Rodríguez, et al. "Mathematical detection of aortic valve opening (B point) in impedance cardiography: A comparison of three popular algorithms." Psychophysiology 54.3 (2017): 350-357.

[2] KELSEY, ROBERT M., et al. “The Ensemble-Averaged Impedance Cardiogram: An Evaluation of Scoring Methods and Interrater Reliability.” Psychophysiology, vol. 35, no. 3, 1998, pp. 337–340., doi:10.1017/S0048577298001310

Thank you for these comments. According to your suggestions, we performed a paired t-test for the values of cardiac output estimated from annotations made by an expert and those found by the algorithm. We added also the value of the Pearson’s correlation coefficient and included the Bland-Altman plot. We discuss these new results in the manuscript.

Round 2

Reviewer 2 Report

I appreciate that the Authors addressed my concerns. I have one final concern regarding the performed paired t-test. In fact, the significance of the t-test implies the presence of a bias of the proposed method in the estimation of the variable with respect to the gold standard. However, based on the Bland-Altman plot and the correlation analysis, it appears that there are no systematic errors and that the values of CO obtained by the two procedures are quite similar (the slope is almost 1 and the constant term is almost 0, which is usually obtained for good models). Please confirm that the t-test is not significant or discuss the possibility of correcting the method by adding the bias (i.e., the difference between the two procedures' mean values) to the values obtained by the tested method.

Author Response

Dear Reviewer, 

Thank you for this valuable remark. Indeed, the  t-test was by our mistake (for which we apologize) performed for the related samples of scores. They are of course independent since the indications made by an expert have no influence on the annotations of the algorithm.  We therefore repeated the t—test and found out, this time p-value = 0.4, which indicates that there are no significant differences between assessment made by an expert and the algorithm. We correct the discussion in the manuscript.